# Biventricular Conversion for Hypoplastic Left Heart Variants: An Update

**DOI:** 10.3390/children9050690

**Published:** 2022-05-10

**Authors:** Christopher E. Greenleaf, Jorge D. Salazar

**Affiliations:** Division of Pediatric and Congenital Cardiac Surgery, UT-Houston McGovern Medical School, Children’s Memorial Hermann Hospital, 6431 Fannin Street, MSB 6.264, Houston, TX 77030, USA; jorge.d.salazar@uth.tmc.edu

**Keywords:** functional single ventricle, cardiac surgery, borderline left heart, hypoplastic left heart, palliation

## Abstract

Ongoing concerns with single-ventricle palliation morbidity and poor outcomes from primary biventricular strategies for neonates with borderline left heart structures have led some centers to attempt alternative strategies to obviate the need for ultimate Fontan palliation and limit the risk to the child during the vulnerable neonatal period. In certain patients who are traditionally palliated toward single-ventricle circulation, biventricular circulation is possible. This review aims to delineate the current knowledge regarding converting certain patients with borderline left heart structures from single-ventricle palliation toward biventricular circulation.

## 1. Introduction

Ongoing concerns with single-ventricle palliation morbidity and poor outcomes from primary biventricular strategies for neonates with borderline left heart structures have led some centers to attempt alternative strategies to obviate the need for ultimate Fontan palliation. The objective of this manuscript is to review the current literature on gaining successful biventricular circulation in patients with borderline left heart structures with patency of the aortic and mitral valves.

## 2. Definition of Borderline Left Heart Structures

A satisfactory definition of what constitutes “borderline left heart structures” is complicated by the fact that it is a diagnosis of exclusion. There is universal agreement about some amalgamations of morphologies that are biventricular, i.e., bicuspid aortic valve without stenosis and coarctation, or univentricular, i.e., HLHS with aortic and mitral atresia. Large portions of the diagnoses along the continuum between these two morphologies are managed very differently amongst centers. Part of the issue is that the definition of biventricular is a matter of the success of the postoperative functional status of repair, depending on both ventricles providing adequate cardiac output with low left atrial pressures, for which we have limited preoperative predictors. Because of the potential dire consequences of choosing a biventricular strategy and failing [1], reasonably, many centers have chosen to take most “borderline” cases down the univentricular palliation pathway, as it has a confirmed set of outcomes and risks.

This difficulty in the inter-center prediction of biventricular success is evident in the poor external validity of scoring systems, such as the Rhodes score [2] and the CHSS score [3]. In a study by Tami et al. [4], the neonates in their study had a left ventricular adequacy score that would have predicted a high risk of mortality with biventricular repair. There were no early or late deaths at a follow-up of 38 ± 16 months after the operation, and only 4 patients (20%) had mild symptoms. The rest of the patients were asymptomatic.

The Rhodes and CHSS scores were specifically validated for patients with critical aortic stenosis. Other anatomic substrates of hypoplastic left heart variants, such as unbalanced atrioventricular septal defects (uAVSD), posterior mal-aligned ventricular septal defects, etc., may not be appropriate to utilize these scores. Since the early 2000s, there have been multiple studies utilizing echocardiography to predict the postnatal surgical outcome. In a retrospective study of fetal echocardiographic images by Jantzen et al. [5], researchers attempted to use such fetal images to distinguish postnatal biventricular vs. univentricular surgical management. The fetal echocardiographic parameters that best distinguished between the two surgical strategies included mitral valve annulus z-score (univentricular −3.3 ± 1 vs. biventricular −2.6 ± 1.4, *p* = 0.048), left ventricular end-diastolic dimension z-score (univentricular −3.3 ± 1 vs. biventricular −2.6 ± 1, *p* = 0.04), aortic valve annular dimension z-score (univentricular −3.1 ± 1 vs. biventricular −2.4 ± 0.94, *p* = 0.04), and ascending aortic diameter z-score (univentricular −3.2 ± 0.7 vs. biventricular −2.6 ± 0.9, *p* = 0.02). A mitral valve z-score ≥−1.9 or a tricuspid: mitral valve ratio ≤1.5 suggests a high probability of biventricular repair; whereas, a right: left ventricular end-diastolic dimension ratio ≥2.1 confers a likelihood of single-ventricle palliation.

In a study to identify preoperative parameters that predict intervention-free survival with biventricular circulation after primary aortic arch repair [6], infants who underwent aortic arch repair, with an aortic and/or mitral valve z-score ≤−2, were analyzed. A total of 51 out of 73 patients (70%) were living with biventricular circulation and had no re-interventions in a year. In a classification and regression tree analysis, a mitral valve to tricuspid valve ratio ≤0.66, with an aortic valve z-score ≤−3, had the greatest power to predict biventricular failure (sensitivity 71% and specificity 94%). When surgical papers about biventricular conversion or left ventricular recruitment describe their inclusion criteria, they typically use some combination of left-sided valvar or left ventricular end-diastolic dimension z-scores of ≤−3 as a cut off for inclusion.

Mart et al. [7] created a scoring system that better discriminates, than the Rhodes or CHSS score, in predicting biventricular repair in a cohort of neonates with hypoplastic left heart complex. The formula includes the mitral valve: aortic valve annulus ratio divided by the left ventricular: right ventricular length ratio, then adding the main pulmonary artery diameter, indexed to body surface area. Using a cutoff value of ≤16.2, biventricular repair would have been predicted with a sensitivity, specificity, positive predictive value, and negative predictive value of 1.0. This 2V-Score needs prospective, multi-center validation.

Differentiating borderline left heart structures is still an imprecise science. Advanced imaging modalities have the potential to add additional data to help with decision making. Computed tomography (CT) can add to the precision of the assessment of ventricular columns and morphometric parameters [8]. Several studies have identified a left ventricular end-diastolic volume between 20 and 50 mL/m^2^ as potentially adequate ventricles to maintain biventricular circulation [9,10,11].

Other anatomies, such as uAVSD, may have straightforward preoperative features to guide the surgical pathway. A multi-institutional study from the CHSS used an echocardiogram to define the limits of uAVSD [12]. An atrioventricular valve index (AVVI), left atrioventricular valve area/total atrioventricular valve area, centimeters squared, was used to discriminate the transition from balanced to unbalanced, and correlate that with the surgical strategy. Patients with an AVVI < 0.19 uniformly underwent univentricular repair. The strategies for patients with borderline left heart structures with an AVVI between 0.19 and 0.39 had very heterogeneous management. Lugones et al. [13] refined an indexed ventricular septal defect measurement to attempt to clarify the borderline patients with 0.19 ≤ AVVI ≤ 0.39 from the CHSS study. This includes the VSD size compared to the atrioventricular valve size to the CHSS AVVI, to distinguish uni- vs. biventricular surgical strategies.

Unfortunately, despite the advances in echocardiographic imaging, these modalities frequently have inadequate sensitivity. We have found that before committing a child to single-ventricle palliation, it is frequently prudent to perform intracardiac exploration. We and other groups have found that, frequently, preoperative imaging may underestimate the actual size of structures, especially when the subvalvar mitral apparatus is normal [14].

## 3. Left Ventricular Recruitment

There is a concern that patients with borderline left heart structures, who are forced into biventricular circulation before they are ready, may have left atrial hypertension and poor cardiac output that can lead to death or to poor future candidacy for univentricular conversion or heart transplant [1]. For these patients, a strategy was devised to harness the heart’s intrinsic growth potential toward a more adequate left heart by “force volume loading” the left side using an approach they termed “staged left ventricular recruitment.” [15]. This can include endocardial fibroelastosis resection, left-sided valvuloplasty, atrial septal defect restriction, and the augmentation of pulmonary blood flow. Thirty-four patients with borderline left heart structures, defined as left-sided heart structure z-scores between −5 and −0.5, with aortic and mitral valve patency, were compared to similar patients who had undergone traditional single-ventricle palliation. Atrial septal defect restriction was determined by the presence of a transseptal gradient >5 mmHg, which was usually achieved by restricting the atrial septum down to an approximately 4 mm opening. The determination of how restrictive to make the atrial septum is performed with consideration of the mitral valve size and the left ventricular size. In an attempt to increase left ventricular preload, a systemic to pulmonary artery shunt or a right ventricular to pulmonary artery (RVPA) conduit is added. Figure 1 is a pictorial description and intraoperative picture of a patient undergoing LV recruitment with an RVPA conduit.

This was an asymmetric cohort study with a mix of historical and contemporary controls, leading to the possibility of some selection bias. In fact, the staged LV recruitment cohort had a trend toward a larger left ventricular end-diastolic volume (LVEDV) z-score (−2.5 ± 1.2 vs. −2.9 ± 0.2, *p* = 0.07), and they were more likely to have had a balloon aortic valvuloplasty at some point, preoperatively. Twelve of the nineteen patients (63%) with significant ASD restriction would receive catheter-based balloon dilation or stenting for significant left atrial hypertension. As a recurring theme that will be observed consistently in all of these series of left ventricular recruitment and biventricular conversions, there was a need for more reinterventions with left ventricular recruitment and a longer length of hospital stay. The median cumulative duration of all hospitalizations from stage 1 single-ventricle palliation to the most recent follow-up was 94 days (range 37 to 518 days) in the left ventricular recruitment group, and 54.5 days (range 8 to 348 days) in the single-ventricle palliation group (*p* = 0.006). The wide range underscores how these patients can differ among themselves, and stresses, once more, the need for individualized, personalized care.

Luckily, there is an overall increase in the ejection fraction between the postnatal echocardiogram and the echocardiogram before biventricular conversion (*p* = 0.004). There was improved freedom from death or transplant with the left ventricular recruitment strategy over single-ventricle palliation (88.2% vs. 76.5%, respectively). Thirteen patients (38%) achieved biventricular circulation after left ventricular recruitment. At catheterization, before biventricular conversion, the mean left atrial pressure, with balloon occlusion of the atrial septal defect, was 14 ± 4.7 mmHg. At a median follow-up of 2.9 years (range 1 to 6 years), all of these patients were alive. As a proof of concept, this manuscript demonstrated significant success. The LVEDV and left ventricular long-axis z-scores were not significantly different between groups before stage 1 palliation, but there were significant increases in size in the left ventricular recruitment group before bidirectional Glenn (*p* < 0.005) and before Fontan or biventricular repair (*p* < 0.01). Of note, restriction of the atrial septum was the only predictor of an increase in LVEDV (*p* < 0.001). It is possible to increase the left heart dimensions by using a left ventricular recruitment strategy, by “force volume loading” the left heart in a stepwise manner.

Figure 2 shows the enlargement of the left ventricular volume, from before left ventricular recruitment and after full biventricular conversion, in one of the patients at our center who underwent the staged ventricular recruitment pathway. Note the interventricular septal shift that leads to near-normal relative ventricular volumes. It seems that there is more change than just increased preload leading to septal shift alone. In some studies, the increase in left ventricular dimensions is not associated with a simultaneous decrease in the size of the right ventricle [16]. On top of the volume increases, the indexed left ventricular mass increased, and there was a decrease in the left ventricular mass–volume ratio, since the ventricle responds to volume, as well as to pressure load. The Boston left ventricular recruitment study excluded patients with ventricular septal defects (VSDs). The group at Chiba Children Hospital in Japan discussed their staged biventricular repair-oriented strategy in borderline candidates with VSDs [17]. Importantly, recognizing the limitations of the strategy, they chose patients to undergo biventricular repair by using a risk profile for single-ventricle palliation, rather than by a purely anatomical possibility. They were more likely to follow the staged biventricular repair strategy if there were features that had the potential for increased pulmonary hypertension, i.e., Kartagener syndrome or Trisomy [18]. This was an approach reinforced by Boston, with the evolution of their biventricular repair and left ventricular recruitment program [19]. They performed a study to determine the effect of ASD restriction without VSD closure on ventricular growth in patients with borderline right or left ventricles and VSDs. There were increases in the median indexed ventricular diastolic volume (31.7 mL/m [IQR, 24.5–37.1] to 48.5 mL/m^2^ [IQR, 38.4–58.0]; *p* < 0.01) and median indexed systolic volume (13.3 mL/m^2^ [IQR, 9.7–18.7] to 19.5 mL/m^2^ [IQR, 16.8–29.7]; *p* < 0.01). Biventricular conversion was performed in 14 patients (67%), with 2 deaths (14%).

Along with staged ventricular recruitment, initial palliation with a hybrid Norwood and keeping the atrial septal defect somewhat restrictive has also led to adequate growth in the left heart structures. Yerebakan et al. [20] and the team in Giessen, Germany, described their patients with hypoplastic left heart variants who underwent hybrid Norwood, and then eventual biventricular repair. Similar to staged ventricular recruitment, the median aortic valve z-scores (−2.13 [range, −7.14 to 0.77] to −1.21 [range, −8.87 to −0.08]; *p* = 0.04) and mitral valve z-scores (0.52 [range, −2.97 to 2.42] to 0.98 [range, −2.35 to 2.64]; *p* = 0.05) increased after the hybrid Norwood procedure. They did not report LV length z-scores, but there was no change in the LV/RV ratio. Of interest, in looking toward facilitating LV growth in these patients, they maintained a restrictive interatrial communication. Left atrial pressure up to 15 mmHg and a gradient across the ASD between 5 and 10 mmHg seemed to be tolerable. The patients in this study were included if they underwent a hybrid Norwood and biventricular repair. No comparison could be made with patients who had the anatomy for potential biventricular repair, but who were unable to achieve adequate left-sided growth to attempt biventricular repair. Furthermore, other centers have shown the potential for left heart growth after the hybrid Norwood procedure [18,21,22,23]. In yet unpublished results at our center, we have found excellent growth of the left heart with LV recruitment after single-ventricle palliation. The median LV long-axis z-score increased from −4.65 (−4.9 to −3.4) to −1.75 (−4.7 to −1.3) after LV recruitment (*p* < 0.05) (Figure 3).

### Surgical Techniques

The valvar and left ventricular outflow tract operative techniques are similar to other strategies performed on pediatric patients. As observed above, restriction of the atrial septal defect is consistently the most significant maneuver to help grow the left-sided cardiac structures. A patch with a fenestration in it or a partial primary closure can be used. The residual shunt has been anywhere between 2 and 6 mm, depending on the perceived sizes of the mitral valve, aortic valve, and left ventricle. Our center and other centers use judicious balloon septostomy as needed, to keep the left atrial pressure ≤15 mmHg. To further strengthen the argument, inter-atrial communication that is left open at the time of aortic arch augmentation in neonates with biventricular physiology is predictive of a need for reintervention on the left ventricular outflow tract [24].

The other key surgical piece of left ventricular recruitment is the augmentation of pulmonary blood flow. A Sano or modified Blalock-Taussig shunt are employed in a standard fashion. If there is a bidirectional Glenn (BDG) present, then we may choose to place a pulmonary artery band between the shunt and BDG to prevent the reversal of flow in the superior vena cava. It is not at all evident that augmentation adds to left heart growth. It makes intuitive sense, but, in fact, Qp and Qp: Qs were not found to be statistically significant predictors in the development of left heart growth [19]. The need for a shunt >6 mm was predictive of heart transplant or death after left ventricular recruitment. Despite this, we are not aware of any center, including ours, that does not add more pulmonary blood flow during left ventricular recruitment. Many of the patients have a BDG and are around the age that they would undergo Fontan. They would probably become too cyanotic to tolerate waiting the additional 6–12 months, without the added pulmonary blood flow before biventricular conversion.

## 4. Biventricular Conversion

Six-to-twelve months after left ventricular recruitment, when the left heart structures look adequate echocardiographically, we then test the patients’ readiness for biventricular conversion. We obtain an MRI to assess ventricular volumes, valvar morphology, and, if present, ventricular septal defect orientation to the systemic semilunar valve. To assess hemodynamic tolerance, we also obtain cardiac catheterization to assess Qp: Qs and to balloon occlude the atrial septal defect, to check what the left atrial pressures would be after septation with a Qp: Qs near one. If all of this is adequate, then the patient will be recommended for biventricular conversion, after multidisciplinary discussions with pediatric cardiology, pediatric radiology, and pediatric cardiac surgery. Biventricular conversion includes the takedown of aortopulmonary amalgamation; the takedown of a cavopulmonary connection; ventricular septation, if necessary; and separation of the left and right ventricular outflow tracts by direct re-anastomosis, conduit reconstruction, or pulmonary autograft translocation.

Kalish BT et al. [25] described the short-term outcomes of 28 patients with the presence of one or more small left heart structures or significant left ventricular dysfunction, who underwent initial Norwood as palliation and then had biventricular conversion. The patients were reported as HLHS, uCAVSD, and interrupted aortic arch. Risk factors for single-ventricle palliation were present in 14 (50%) patients. Seventeen (61%) patients required re-intervention after biventricular conversion. After biventricular conversion, the LVEDV at echocardiogram increased to 91.33 mL/m^2^ (56.6 to 251.5 mL/m^2^) from 58.1 mL/m^2^ (26.6 to 97.5 mL/m^2^) in the HLHS group, and to 58.5 mL/m^2^ (45.7 to 69 mL/m^2^) from 28.1 mL/m^2^ (16 to 86.2 mL/m^2^) in the uCAVSD group; *p* < 0.05 in both groups. The left ventricular end-diastolic pressure (LVEDP) increased to 17 mm Hg (9 to 29 mm Hg) from 12 mm Hg (6 to 20 mm Hg) in the HLHS group; *p* < 0.05. The LVEDP non-significantly increased to 11 mm Hg (8 to 24 mm Hg) from 8 mm Hg (7 to 10 mm Hg) in the uCAVSD group; *p* = 0.079.

Further investigation of preoperative parameters and postoperative outcomes in patients with HLHS or uCAVSD, undergoing biventricular conversion, was performed [26]. Multivariable Cox regression showed that LVEDP ≥ 13 mm Hg (adjusted hazard ratio, 4.00; *p* = 0.037) and postoperative right ventricular pressure > ¾ of the systemic pressure (adjusted hazard ratio, 21.75; *p* < 0.001) were significantly associated with the primary composite endpoint of death, heart transplant, or biventricular repair takedown. Of 51 patients, 11 (22%) experienced the primary endpoint. Patients with HLHS were more likely to experience this endpoint compared to those with uCAVSD (30% vs. 6%, *p* = 0.03). A total of 25 (49%) patients required surgical re-intervention after biventricular conversion. There was a transplant-free survival of 80% at 3 years. The survival rate was 85% for the group with LVEDP < 13 mm Hg and 60% for the group with LVEDP ≥ 13 mm Hg (*p* = 0.037). This survival rate does not include any patients who were censored before they were deemed adequate biventricular conversion candidates. Considering the survival and reoperation rates, the question arises as to whether a focus on patient selection for those patients with risk factors for single-ventricle palliation, or outright single-ventricle palliation failure, should trump strict anatomic and functional criteria [27].

To help address the concerns with this question, Boston looked specifically at patients with uAVSD [28]. These patients were divided into three groups. The groups were those who had undergone single-ventricle palliation, those who underwent primary or staged biventricular repair, and those who underwent biventricular conversion from single-ventricle palliation. The median length of follow-up was 35 months (range 1–192 months). In the multivariable analysis, the single-ventricle palliation and biventricular conversion groups had a higher risk for catheter-based re-interventions, and this was worse in heterotaxy patients. There was a survival advantage, by Kaplan–Meier estimates, with the biventricular and biventricular conversion/biventricular repair group compared to the single-ventricle palliation group (log-rank *p* = 0.005).

As for patient selection, in 23 patients who underwent biventricular conversion for Fontan failure, the 2-year survival rate was 72.7% (95% confidence interval, 37–90%) [29]. This is similar to orthotopic heart transplant outcomes in patients for failing Fontan. All the elective Fontan takedown patients survived, with a median follow-up of 1.1 years (IQR, 0.2 to 2 years). Long-term follow-up will be important in biventricular conversion patients, because even after the LVEDV z-scores reached normal or elevated ranges, diastolic dysfunction was common and progressive. Median LVEDP increased from 12 mm Hg before biventricular conversion to 22 mm Hg at the last follow-up. We found similar significant LVEDP increases in our biventricular conversion patients, with the median LVEDP increasing from 5.5 mm Hg (4 to 10 mm Hg) to 10 mm Hg (6 to 20 mm Hg) [30]. In the six patients for whom the right ventricular pressure (RVP) could be estimated by a tricuspid valve jet, the RVP was less than half the systemic pressure in all of those patients. One or more risk factors for single-ventricle palliation were present in three (23%) patients. There was a 92% survival rate at a median follow-up of 22.6 months (range, 0.3 to 36.4 months). Our patients were carefully selected and included both patients with Fontan failure and with well-functioning single-ventricle palliation. This may explain the improved survival over the biventricular conversions for purely Fontan failure described above.

## 5. Conclusions

In certain patients who are traditionally palliated toward single-ventricle circulation, biventricular circulation is possible. The short- and medium-term results are satisfactory. This circulation offers the prospect to prevent the well-known Fontan morbidities at the consequence of greater re-interventions, left atrial hypertension, and unknown long-term outcomes. Further studies and follow-ups are needed to assist in refining optimal patient selection, improved surgical techniques, and proper postoperative management.

## Figures and Tables

**Figure 1 children-09-00690-f001:**
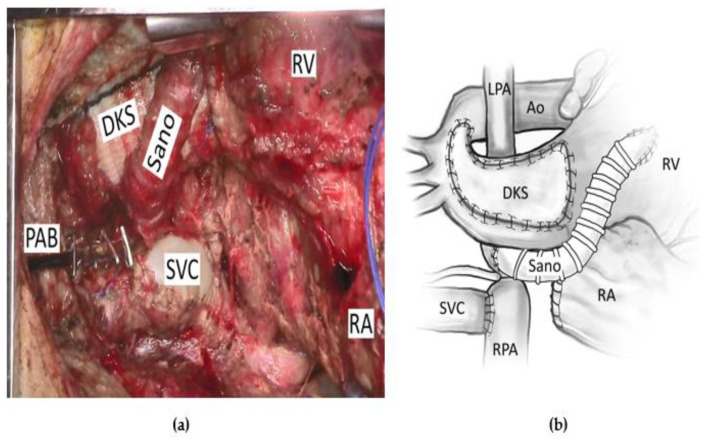
A patient undergoing a left ventricular recruitment procedure with (**a**) an operative picture and (**b**) a pictorial description of the same operation. Ao = aorta, DKS = Damus–Kaye–Stansel anastomosis, LPA = left pulmonary artery, PAB = pulmonary artery band, RA = right atrium, RPA = right pulmonary artery, RV = right ventricle, SVC = superior vena cava.

**Figure 2 children-09-00690-f002:**
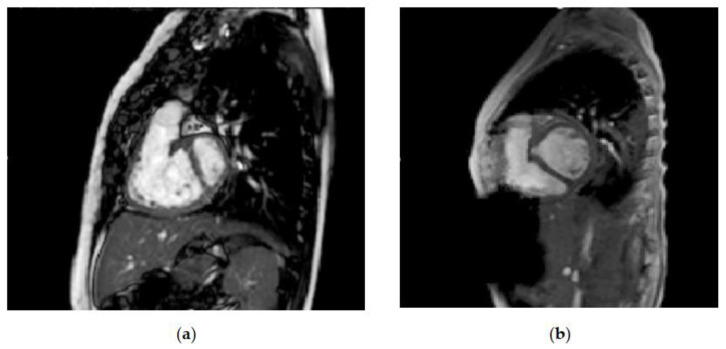
Comparison of left ventricular volumes on magnetic resonance imaging (**a**) before left ventricular recruitment: left ventricular end-diastolic volume (LVEDV) 12.9 mL, indexed left ventricular end-diastolic volume (LVEDVi) 20.2 mL/m^2^ and (**b**) after full biventricular conversion: LVEDV 54 mL and LVEDVi 72 mL/m^2^.

**Figure 3 children-09-00690-f003:**
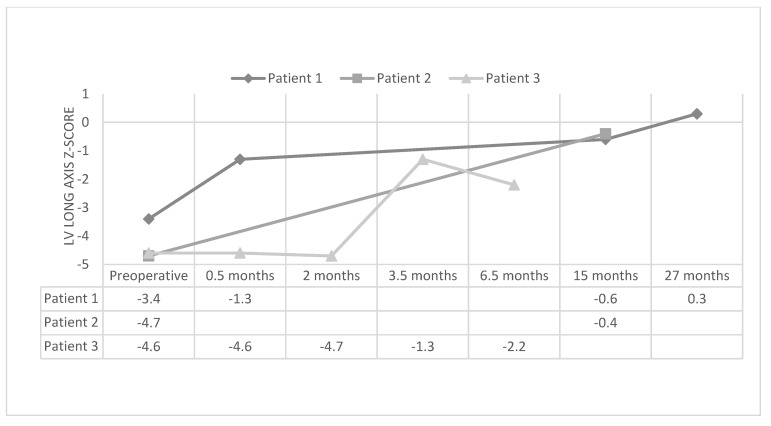
Left ventricular long-axis z-score in patients before and after left ventricular recruitment.

## Data Availability

No new data were created or analyzed in this study. Data sharing is not applicable to this article.

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
