# Peer review of "Biventricular Conversion for Hypoplastic Left Heart Variants: An Update"

_children, 2022, doi:10.3390/children9050690_

Round 1
Reviewer 1 Report
The authors reviewed the information of the biventricular conversion for hypoplastic left heart variants. The contents are meaningful and clinically important. However, there are some issues with the manuscript for the authors to concern in the revision.
1. The advance of the mechanisms for fetal development of small left heart structures should be introduced more clearly in a table.
2. A table showing the advance of the biventricular conversion studies or practice should be provided.
3. The authors are encouraged to update the 2022 references if possible.
4. The pediatric cardiac surgeons are needed to review this manuscript.
Author Response
The advance of the mechanisms for fetal development of small left heart structures should be introduced more clearly in a table.
This section has been removed as per the academic editor's comments
A table showing the advance of the biventricular conversion studies or practice should be provided.
If insisted, I would add this, otherwise I would prefer not to make this change
The authors are encouraged to update the 2022 references if possible.
The references are updated though March of 2022
Reviewer 2 Report
Excellent and comprehensive review of strategies to recruit the borderline heart to a BiV circulation.
the references are excellent and the figures are a helpful addition to the well-written manuscript
Author Response
Excellent and comprehensive review of strategies to recruit the borderline heart to a BiV circulation.
the references are excellent and the figures are a helpful addition to the well-written manuscript
Thank you. Much appreciated
Reviewer 3 Report
A well-prepared publication. It concerns a very interesting topic. The authors present an overview of the publications and their own experience in this topic. The work is of clinical importance for patients with HLHS. Line 307 is some mistake "AND" is writen in capital leter.
Author Response
A well-prepared publication. It concerns a very interesting topic. The authors present an overview of the publications and their own experience in this topic. The work is of clinical importance for patients with HLHS. Line 307 is some mistake "AND" is writen in capital leter.
Fixed